

# Stochastic Hydrogeology's Biggest Hurdles Analyzed and Its Big Blind Spot

Yoram Rubin[1], Ching-Fu Chang[1], Jiancong Chen[1], Karina Cucchi[1], Bradley Harken[1], Falk Heße[2], Heather Savoy[1]

[1]Civil and Environmental Engineering, University of California, Berkeley, 94720, USA
[2]Computational Hydrosystems, Helmholtz Centre for Environmental Research (UFZ), Leipzig, 04318, Germany

*Correspondence to:* Yoram Rubin (rubin@ce.berkeley.edu)

**Abstract.** This paper considers questions related to the adoption of stochastic methods in hydrogeology. It looks at factors affecting the adoption of stochastic methods including environmental regulations, financial incentives, higher education, and

the collective feedback loop involving these factors. We begin by evaluating two previous paper series appearing in the stochastic hydrogeology literature, one in 2004 and one in 2016, and identifying the current thinking on the topic, including the perceived data needs of stochastic methods, the attitude in regulations and the court system regarding stochastic methods, education of the workforce and the availability of software tools needed for implementing stochastic methods in practice. Comparing the state of adoption in hydrogeology to petroleum reservoir engineering allowed us to identify quantitative

metrics on which to base our analysis. For impediments to the adoption of stochastic hydrology, we identified external factors as well as self-inflicted wounds. What emerges is a picture much broader than current views. Financial incentives and regulations play a major role in stalling adoption. Stochastic Hydrology's blind spot is in confusing between risk and uncertainty and ignoring uncertainty. We show that stochastic hydrogeology comfortably focused on risk while ignoring uncertainty, to its own detriment and to the detriment of its potential clients. The imbalance between the treatment on risk on

one hand and uncertainty on the other is shown to be common to multiple disciplines in hydrology that interface with risk and uncertainty.

## 1 Introduction

A key element in the discussion on adoption of stochastic hydrogeology (SH), or the lack of it, is the apparent disconnect between theory and practice. The root causes for the perceived failure in adoption have been discussed in several papers

(published in 2004 and 2016; for a complete list see Section 2). In this study, we are set to take yet another look at this issue and draw some conclusions based on a factual basis. Regardless of whether true or not, this apparent gap raises questions such as: First, what does 'practical' mean: does it mean "has no use in real-life" or "useful but hard to apply, and would you please simplify it and make it more accessible"? How real the perceived gap is and how much of it depends on the skills of





the commentator? Is there a reasonable delineation between theory and practice? For example, would it take a mobile app with all-inclusive user-interfaces in order for SH to be viewed as practical? These and other questions will be addressed here.

There is an important issue looming in the background of this conversation. SH intends to model uncertainty, and uncertainty exists and must be dealt with, regardless of any perceived shortcoming. One can argue about strengths and weaknesses of

SH, e.g. the practicality, but there is no argument about the need to address uncertainty. But who is to blame for ignoring uncertainty? Are there some external factors not related to the science? And, what has SH missed, if anything, that could have made a difference in adoption? This paper reviews the evolution of the thoughts on these topics. It evaluates the strengths and weaknesses of arguments made and provides additional perspectives by looking at factors not considered before.

The motivation for this paper originated from the need to get to the root causes of some of the issues that we view as critical and from the growing recognition that there is a need to both go beyond the scientific and technical issues associated with SH and to consider issues related to education, public policy, economics, and the law. However, this paper is not just about external factors: we will also look at SH's biggest omissions.

The remainder of the paper is organized as follows: Sect. 2 summarizes and compares the 2004 SERRA (*Stochastic*

*Environmental Research and Risk Assessment*) and 2016 WRR (*Water Resources Research*) series; Sect. 3 reviews available SH-related software; Sect. 4 compares the role of stochastic methods in hydrogeology and petroleum engineering; and Sect. 5 discusses legal and regulatory perspectives about stochastic methods; Sect. 6 discusses the perception of uncertainty broadly and in SH; and Sect. 7 concludes with recommendations and outlook for the field of SH in practice.

## 2 Summary of perspectives from the 2004 and 2016 paper series

To begin the SH-specific discussion, we provide a brief summary of the comments made in two series of articles addressing SH in practice. These series were published in SERRA in 2004 and in WRR in 2016. In our subsequent discussion we shall refer to the '2004 series' (Christakos, 2004; Dagan, 2004; Freeze, 2004; Ginn, 2004; Molz, 2004; Neuman, 2004; Rubin, 2004; Sudicky, 2004; Winter, 2004; Zhang and Zhang, 2004) and to the '2016 series' (Cirpka and Valocchi, 2016; Fiori et al., 2016; Fogg and Zhang, 2016; Rajaram, 2016; Sanchez-Vila et al., 2016). The 2004 series asked nine researchers the

following two specific questions: (1) 'Why have there not been many real-world applications of stochastic theories and approaches, despite the significant progress in developing such rigorous theories and approaches for studying fluid flow and solute transport in heterogeneous media?', and (2) 'In your opinion, what must be done in order to render stochastic theories and approaches as routine tools in hydrogeologic investigation and modeling?'. The 2016 series is more open-ended, asking four teams of researchers to debate on 'stochastic subsurface hydrology from theory to practice'. We chose these two series

as a baseline for discussion because they address similar questions, they combine multiple perspectives from a variety of



researchers, and the 12-year timespan between them allows us to consider the evolution of SH. However, due to the more open nature of the 2016 series, not all of the discussion points can be compared between the two series. Of note is that we refrain from considering issues related to transport modeling specifics in as much detail as the 2016 WRR series due to the wider scope of our questions regarding stochastic methods in general.

## 2.1 A brief summary of the issues

The themes pursued by both paper series in trying to explore the gap between academia and practitioners regarding the adoption of SH are as follows:

**1. Regulations and litigation**: There is a broad agreement (but not a consensus) that institutional attitudes (in the form of regulations, public policy and the legal system) are major factors to consider, and that in the case of SH, the institutional attitudes have played a negative role. In a departure from this broadly-held view, Christakos (2004) opines that no blame should be assigned on the court system when other fields use models with uncertainty, e.g. biomedical models for DNA analysis, under legal scrutiny. **Questions we shall address:** What is the evidence to support the claim about institutional resistance? Is there evidence to the contrary?

**2. The role of higher education**: Authors criticized the lack of academic preparation in the theoretical fundamentals of stochastic theory. Suggestions were made that SH theory is too complex to be taught. **Questions we shall address:** What is the factual basis for this statement? Is there evidence to the contrary? Who drives the development of teaching curricula, and what can be learned about it from other disciplines?

**3. Lack of appropriate measurement technology and/or data**: A major theme in the 2004 series is the apparent lack of technologies needed for providing measurements attributable to the parameters and scales of stochastic models. Additionally, it is suggested that the application of SH requires large amounts of data. **Questions we shall address:** What are the elements of SH that require lots of measurements? Are there statistical tools to handle situations with severe shortage of measurements?

**4. Lack of user-friendly software that applies theory:** This topic mainly appears as the call for user-friendly software that integrates multiple forms of information and that is computationally efficient enough for practitioners to use. **Questions we shall address**: What is the factual basis for this assertion?

**5. SH theory is out of touch with reality:** Arguments here were either directed to the theoretical research as not applicable in real-world problems (e.g. oversimplification of spatial structure, or minimizing uncertainty in parameters when other uncertainties need to be minimized), and to the lack of applications to showcase the applicability of theories in real-world problems. **Questions we shall address**: Is there a factual basis for these arguments? Is there any particular aspects of SH that gives rise to such arguments?



The topics most prevalently mentioned in the 2016 WRR series are Topic 4 (user-friendliness of concepts) and Topic 5 (theory too limited or not showcased in applications). Concerns regarding the availability of useful software tools remain (Cirpka and Valocchi, 2016), however there is some recognition that few stochastic modules were introduced into forward models (Fiori et al., 2016).

## 2.2 Possible misconceptions and general observations

In assessing the root causes of the so-called gap, we need to distinguish between real and perceived obstacles. Perceived obstacles reside in myths and misconceptions. Real obstacles are everything else.

It is often stated that SH requires lots of data for real-life applications. As is the case with most persistent myths, there is a grain of truth here, and this is the observation that getting a well-defined variogram (or for that matter, other SRF models such as multi-point spatial statistics) requires dozens of measurements. Whereas this observation is true, everything else that is added on top of it is not true. The variogram is a stochastic method for representing the spatial variability structural patterns with a model that is parsimonious compared to a field of unknown values but allows for the uncertainty in the local variability in un-sampled locations. A shortage of field data does make accurate modeling of variograms difficult, but that is not a gap in SH. In fact, SH is amply equipped to dealing with that challenge. For example, instead of wrongly using (and wrongly assuming) the existence of a deterministic variogram model, one can consider simultaneously several potential variogram models instead of only one. Additionally, one can introduce parametric uncertainty into the estimation process (Rubin, 2003).

Fiori et al. (2016) noted the significant advancement of measurement technologies. And while this is commendable and most welcome, it does not appear to have made a significant change in closing the gap. This suggests that measurement technologies, although they advance the field of hydrogeology, should not be viewed as a critical factor in closing the gap for SH. SH's primary goal is modeling of uncertainty, and uncertainty exists (and must be modeled) regardless of limited availability of measurements and the limitations of measurement technologies. The data helps in reducing uncertainty, but it cannot make it disappear (at least not in field applications). The need for large amount of data in this context is somewhat of an oxymoron: having lots of data is useful, but lack of data does not make uncertainty go away and in fact it only accentuates the need for stochastic methods. Should theories be accepted or rejected based on the availability of appropriate measurement technologies? If that were true, then the theory of relativity would have been kept in the drawer until the 1970s.

As noted by Cirpka and Valocchi (2016) and Li et al. (2018), Bayesian methods can use a variety of forms of information including prior knowledge and indirect data to help constrain the uncertainty in these parameters. Variograms can be used even in the absence of a single measurement based on prior information. This would result in large uncertainty in the predictions, but (1) that is not a problem, but just a reflection of reality, and (2) that is the role of SH: to predict uncertainty,



not to make it go away. The option of giving up on modeling uncertainty on the grounds that there is not enough data is not justified.

Moving forward, we will refrain from focusing on data needs based on our understanding that scarcity of data does not contribute to the gap. For the same reason, we will also avoid discussing the (always welcome) effort towards improving measurement technologies. Rather, we should focus on issues 1, 2 and 4. We will show that there is a factual base that allows evaluating the veracity of these issues. We will also address what we call the big miss of SH.

## 3 Software issues

The 2004 and 2016 paper series both identified the lack of SH software solutions as a critical hindrance on the road to acceptance of stochastic methods by practitioners. Neuman (2004) mentioned the lack of ready-to-use and user-friendly software packages. Dagan (2004) and Winter (2004) suggested that most software solutions tend to over-simplify hydrogeologic complexity and thus are of limited appeal. Cirpka and Valocchi (2016) pointed out that stochastic software solutions for reactive transport are in their infancy. Fiori et al. (2016) mentioned that "Practical tools guided by the stochastic paradigm are very few, although the last few years have seen a promising increase of them". Here, we wish to explore the software aspect of SH further through a limited survey of the software landscape. Toward this goal, we developed a list of stochastic software solutions and related them to various components of SH studies. We investigated the extent to which these solutions are capable of handling a variety of SH problems. We do not intend to provide here an exhaustive summary of all SH software tools. Our plan is to identify whether (or not) a critical mass of software tools exists, such that most geologies and most aspects of SH studies could be addressed using an existing software solution.

Critical mass for software is a debatable concept. If one defines critical mass as having readily available software for all situations, critical mass would forever stay an elusive goal. A less demanding and possibly more realistic definition would be to consider readily available solutions as well as source codes that could be modified to address special situations. There is an ever-increasing library of open-source software solutions and modular, adjustable solutions requiring minimal programming skills for applications and it makes significant contributions to the critical mass. Hence, we will avoid making calls about whether or not critical mass is available, but we will show that we are possibly very close and way beyond the common perception.

### 3.1 Classification of stochastic software with examples

There is a wide range of software solutions relating to SH that we could mention. In this study, we focused on software solutions consistent with the definition of SH given in the introduction. We therefore limit our search to software solutions that are guided by complete statistical theories or that make substantial use of statistical concepts.



The three application areas covered in our review include: (1) simulation of random fields, (2) stochastic forward modeling, and (3) stochastic inverse modeling. The first area focuses on the simulation of spatially variable flow and transport domains using geostatistical models. The second area focuses on simulating dependent hydrogeological variables, such as pressure or concentration, in spatially variable domains. The third area focuses on identifying the space random function (SRF) models that could then be used in application under areas 1 and 2.

Table 1 provides a list of software packages and their primary focus area. Our selection does not imply best of class and should not be viewed as a recommendation. We recognize that forward and inverse modeling tools utilize (and may even have built-in) random field generators (RFGs), but in general, random field generation is not their primary focus. Hence, we identified each software package only by its primary application area(s).

**3.1.1 Random field generation and data exploration**

The first application area refers to the generation of spatially variable fields and field data exploration. Available software packages cover a wide range of SRF models that can be, and have been, used to characterize a diverse range of geological environments, from unlithified sediments to sedimentary bedrock aquifers to crystalline bedrocks (see Eaton, 2006) for a more detailed review of type of geologies).

A partial review of the types of SRF models covered by software packages listed in Table 1 is provided in Table 2. In many cases, the SRF models used in the software packages were modified to meet the needs of specific applications, and, with that modification, expanded the range of applicability. For example, when investigating the karst carbonate Biscayne aquifer in southeastern Florida, Sukop and Cunningham (2014) used GSLIB to simulate the spatial distribution of porosity in the carbonated matrix of the karst aquifer after adding capability for simulating nested variogram structures. Castilla-Rho et al. (2014) used Gaussian Random Field Generators (GRFGs) as implemented in SGeMS to simulate small-scale variability in a paleo-valley type aquifer in Northern Chile, where SGeMS is considered as an evolution of GSLIB (Bianchi and Zheng, 2009).

GRFGs represent a class of SRFs useful for modeling continuous variables, such as hydraulic conductivity, due to the parsimony of the Gaussian models. Jankovic et al. (2017) used a three-dimensional version of HYDRO_GEN in order to simulate three-dimensional SRFs of log-normally distributed hydraulic conductivity characterized by exponential and spherical variograms and their influence on breakthrough curves (Bellin and Rubin, 1996). Cvetkovic et al. (1999) used HYDRO_GEN to generate SRFs of a log-normally distributed variable characterized by an exponential correlation structure to model fracture aperture controlling fluid flow and tracer transport in a crystalline rock. While dominant in the early days of SH, GRFGs represent just one of many options currently available.

The development of multi-point statistics (MPS) allowed users to easily simulate non-Gaussian fields (Mariethoz et al., 2010). Although the concept of MPS was explored in the 1990s (Guardiano and Srivastava, 1993), it was Mariethoz et al.'s



(2010) multi-point direct sampling (MPDS) algorithm that opened the door for broad use. Pirot et al.,(2014) used MPDS as implemented in DeeSse to simulate topographies in a braided river system (Straubhaar, 2015). Savoy et al. (2017) used a DeeSse-generated dataset to explore transport through complex geometry that GRFs could not describe.

| Software | Main Focus Area | | | Reference |
| --- | --- | --- | --- | --- |
| | Random Field Generation and Data Exploration | Forward Modeling | Inverse Modeling | |
| DeeSse | ✓ | | | (Straubhaar, 2015) |
| DREAM | | | ✓ | (Vrugt, 2016) |
| FIELDGEN | ✓ | | | (Doherty and Hunt, 2010) |
| GMS | ✓ | ✓ | ✓ | (Aquaveo LLC, 2012) |
| GSLIB | ✓ | | | (Deutsch and Journel, 1998) |
| GSTAT | ✓ | | | (Pebesma, 2004) |
| Isatis | ✓ | | | (Bleines et al., 2004) |
| PEST | | | ✓ | (Doherty, 2005) |
| iTough2 | | | ✓ | (Finsterle, 2011) |
| MAD | | | ✓ | (Osorio-Murillo et al., 2015) |
| SGeMS | | ✓ | ✓ | (Remy et al., 2009) |
| TPROGS | ✓ | | | (Carle, 1999) |
| spMC | | | ✓ | (Sartore, 2013) |
| HYDRO_GEN | ✓ | | | (Bellin and Rubin, 1996) |
| Isim3D | ✓ | | | (Gómez-Hernández and Srivastava, 1990) |

**Table 1: Application areas of reviewed stochastic hydrogeology software**

5    Another method for simulating non-Gaussian geostatistical simulations is the transition probability/Markov chain (TP/MC) approach to the random field generation of categorical variables implemented in TPROGS (Carle, 1999). The TP/MC approach focuses on modeling the spatial arrangement of lithofacies with linkage to basic geological attributes such as volumetric lithofacies proportions, mean lengths, and juxtaposition tendencies. He et al. (2014) used TPROGS to simulate structural heterogeneity for glacial deposits in a headwater catchment in Denmark. Engdahl et al. (2010) used TPROGS to

10   simulate shallow fluvial aquifer that contain representations of sedimentary bounding surfaces. Fleckenstein et al. (2006) used TPROGS to simulate gravel, sand, and muds in the alluvial fan of the Cosumnes River in California, USA.  Indicator



and categorical variables have been implemented in GSLIB for simulating geologic layers (Deutsch and Journel, 1998; Marinoni, 2003). Jackson et al. (2000) used Isim3d to simulate a heterogeneous continuum porous medium to represent fractured rocks (Gómez-Hernández and Srivastava, 1990). In conclusion, a variety of software implementing different theoretical models of spatial variability are available. Our table does not aim to provide a full review of all methods and

associated software for spatial variability, but rather to show that implementations are readily available for multiple types of geologies represented by the different statistical models of spatial variability. We also note that programming capability goes a long way towards expanding the original, stated capabilities of current software packages.

| Software package | Geostatistical models of spatial variability | | | |
|---|---|---|---|---|
| | Gaussian | Non-Gaussian | | |
| | | Multi-Point Statistics | Transition Probability | Indicator/Categorical |
| DeeSse | | ✓ | | ✓ |
| FIELDGEN | ✓ | | | |
| GMS | ✓ | | ✓ | ✓ |
| GSLIB | ✓ | | | ✓ |
| GSTAT | ✓ | | | ✓ |
| Isatis | ✓ | | | |
| SGeMS | ✓ | ✓ | | ✓ |
| TPROGS | | | ✓ | ✓ |
| Isim3d | | | | ✓ |

**Table 2: Popular methods for simulating spatial variability of hydrogeological properties and their implementation in software packages (area 1). Additional options for random field generation are discussed here:**
**http://petrowiki.org/Geostatistical_conditional_simulation#Sequential_simulation.**

### 3.1.2 Forward Modeling

Stochastic software for forward modeling refers to software tools for simulating of hydrogeological processes using spatial distributions of hydrogeological parameters. A forward model, in itself, may not be an SH item, but when coupled with a random field and/or parameter generator, it becomes a bona fide SH application. There are numerous SH studies that couple

process simulators with random field and parameter generators. Most commonly, investigators link between one of the software packages listed above for generating realizations of the SRF with a process simulator of their choice. Several of the software packages listed embed both RFGs and forward models into a single software package. Jones et al. (2005) presented an algorithm implemented within GMS that combines stochastic simulation of hydraulic conductivity using TPROGS with





MODFLOW 2000, a popular numerical model for flow and transport modeling (Harbaugh et al., 2000). Renard and Jeannée (2008) combined geostatistical simulations implemented using Isatis with flow simulations using METIS (Goblet, 1989). Most software packages are able to perform in 2D and 3D. The important point is that the RFGs are standalone and thus can be easily coupled with forward simulators, and this opens the door for a large number of situations amenable to SH treatment.

### 3.1.3 Inverse Modeling

The third application area covers inverse modeling. Inverse modeling is a study in assimilating Type-A (i.e. local data, correlated with a collocated target variable) and Type-B data (non-local, usually related to a subdomain of the target variables). It is intended to produce an image of the subsurface more coherent that what could be achieved with Type-A data alone. As an example, Murakami et al. (2010) used the Method of Anchored Distributions (MAD) algorithm (Rubin et al., 2010) to condition a three-dimensional hydraulic conductivity field at the Hanford 300 Area, Washington, USA. MAD allowed the combination of large-scale injection rate measurements with small-scale electromagnetic borehole flowmeter measurements to derive posterior probability density distributions (pdfs) for geostatistical parameters. In another example, Reeves et al. (2014) combined Type-A data obtained from boreholes with soil deformation data. Chen and Hoversten (2012) and Hou et al. (2006) used seismic AVA and CSEM data to estimate reservoir parameters based on statistical rock-physics models through Markov random fields. Kowalsky et al. (2004) and Kowalsky et al. (2005) combined GPR data (for Type-B data) with Type-A hydrologic data for estimating the flow parameters in the vadose zone. Bellin and Rubin (2004) and Woodbury and Rubin, (2000) used concentration for Type-B data. Chen et al. (2004) used geophysical data for Type-B data for geochemical characterization. Hubbard et al. (1999) used GPR for Type-B data used for identifying the spatial correlation structure. Ezzedine et al. (1999) used geophysical data for Type-B data in a characterization study of a superfund site, and Hubbard et al. (2001) used geophysical data for Type-B data in a characterization study of a large-scale experimental site.

Published inverse modeling software packages cover a wide range of statistical concepts. It is not our goal here to rank and compare inverse modeling strategies, but rather, to provide a glimpse into the diversity for ideas employed and implemented. Detailed reviews are provided in McLaughlin and Townley (1996), Ginn and Cushman (1990) and in Rubin (2003, Chapter 13). There are, of course, different ways to classify inverse modeling concepts, but the most common one is to define the inverse models by the way they treat the hydrological variables. One approach here is to consider the parameters as deterministic yet unknown, which translates into some sort of optimization- or regression-based procedure (e.g. PEST, iTough2, TPROGS, and spMC). The goal of the inverse modeling here would be to define the uncertainty associated with estimating these parameters.



Another approach is to consider the model parameters as random variables, with inverse modeling targeting their pdfs (e.g. MAD and DREAM). Here, we can identify strategies employing Maximum Likelihood (Kitanidis and Lane, 1985; Rubin and Dagan, 1987b), Maximum a-posteriori (McLaughlin and Townley, 1996), and Bayes' Theorem (Osorio-Murillo et al., 2015; Rubin et al., 2010; Vrugt and Ter Braak, 2011). In the Bayesian approach, parameters are described by a pdf, and the

estimation process consists in updating the prior pdf with in-situ measurements. Bayesian methods offer a more complete treatment of uncertainty than optimization methods, as optimization discards non-optimal yet realistic parameter sets, whereas in the Bayesian framework each of the parameter sets is associated with probability and they are all recognized, when assigned with the tiniest probability.

## 3.2 Description of properties of software packages

Several software solutions come with built-in forward model(s) and/or random field generator(s). For example, MODFLOW includes the possibility of importing SRFs simulated by TPROGS (ModelMuse Help, 2009), and the forward model GMS contains modules for the generation of conditional GRF based on FIELDGEN (Aquaveo LLC, 2012).  This approach provides an easy gateway into SH applications, but at the price of limiting the range of applications. Others software solutions are designed to accommodate user-supplied computational modules (e.g. PEST and GSLIB). MAD is a hybrid

between these two approaches, as it comes with built-in modules, but it is also capable of accommodating user-supplied modules. This flexibility opens the door for a wide range of applications, but it could be challenging for users lacking some minimal programming skills.

The modularity of SRF software solutions is a key aspect for their wide adoption. This modularity, implemented for example in GSLIB as a set of Fortran utility routines, allows users to identify specific code sections of relevance and use them in the

data analysis workflow. Other examples of modularity include R packages, such as GSTAT or spMC, which respectively implement parameter estimation based on variograms and transitional probabilities within R (Pebesma, 2004; Sartore, 2013). This allows R users to adopt spatial statistics concepts seamlessly in their workflow. Finally, MAD has been implemented such that it can easily be linked to any RFG or forward model. Even though links to popular models such as MODFLOW and GSTAT are readily available within MAD, using other specific models is made easy thanks to the existence of built-in

links that require minimal coding from the user. Allowing for flexibility in the form of packages that can be integrated into the user's workflow, either as a toolbox within a forward numerical model, or as a package that can be used within a wider programming environment, should facilitate the adoption of stochastic concepts by users.

One final important issue to consider relates to the user-friendliness of software solutions. While some users prefer to work from shell command lines providing flexibility and modularity and allowing the construction of reproducible and automated

workflow, others might prefer a Graphic User Interface (GUI), designed to ease the use of the software by providing user-friendly options for inputting data to the statistical algorithms and visualizing results for interpretation. In our review, all




software provided shell command line versions, thereby allowing practitioners and researchers to randomize parameter field using text input files. On the other hand, GUIs were not available for all packages. For a few geostatistical programs, a GUI was added at later stages of the development as an interface to the geostatistical programs. For example Batgam© is a GUI developed for GSLIB (Buxton et al., 2005). Our review recognizes that different audiences require different capabilities in terms of user-friendliness and therefore reports available forms for using the software solution (Table 3).

## 3.3 Discussion

In this section, we aim to provide further thoughts about other statistical software/toolkits that were not designed directly for SH applications, but could nevertheless be easily connected to SH.  In addition, we will also identify focus areas in need for additional attention and development effort.

Statistical concepts such as Markov chain Monte Carlo simulation, transitional probability, GRF, random walk, etc., have been widely used in SH. In addition to a few of the existing SH packages that have already incorporated those modules, there are other statistical packages that are available, which can be easily linked with hydrogeological models. For example, Nimble provides a way to incorporate Bayesian Hierarchical Modeling into our subsurface parameter estimation (de Valpine et al., 2016) and the R package forecast offers a way to build random walk to help solve reactive transport models (Hyndman and Khandakar, 2008). The geostatistical MATLAB toolbox mGstat provides a MATLAB interface to GSTAT and SGeMS, allowing its users to call GSTAT or SGeMS functions directly from MATLAB and have the output returned seamlessly (Hansen, 2004). These stand-alone statistical packages or software are not designed for SH, but could be easily connected to either existing hydrogeology models or SH toolkits.

There are some areas where important gaps still exist. For example, modeling of uncertainty in karst formations is at its infancy. In another direction, stochastic modeling of reactive transport lags behind modeling of flow processes, mostly due to the complexity of the reaction networks and some ambiguity about the underlying physics. Some of the key transport processes such as diffusion-dominated reactive transport and transverse reactive transport (see the 2016 debate papers, as listed in Section 2) are still debatable. The challenge here, it might be argued, is not in the SH side. Once a clearer picture of the physics and chemistry in complex systems would emerge, the corresponding stochastic software could be used to account for uncertainties due to spatial variability. We, in fact, disagree with such an argument, because uncertainty must be dealt with whether or not an underlying physical model does exist. We will expand on that in a section devoted to the shortcomings of SH.

In summary, a wide variety of stochastic software packages and toolkits are available for practitioners and researchers and have proven capability for modeling and analyzing a wide range of conditions. Despite significant efforts in linking between the software packages representing the various aspects of SH, we note that in many cases some minimal programming skills are needed in order to implement numerical solutions targeted at the specific needs of the modeling exercise. One could





always aspire for seamless links between all packages, but the need of programming skills is not a credible reason for explaining why practitioners in environmental engineering and groundwater hydrology are ignoring opportunities offered by the stochastic approach. We propose that concerns about limitations of software solutions may be rooted in having (or not having) programming skills. It is hard to believe that a sophisticated application in petroleum reservoirs would be scuttled

5   because of difficulties in linking software packages, and so, it is not the lack of programming skills but the availability to pay for programming skills. And for this, you must have clients who are willing to pay. We will explore this question in the subsequent section.

| Software | User-friendliness | | Open Source | Free/commercial | |
|---|---|---|---|---|---|
| | GUI | Third-party GUI | | Free | Commercial |
| DeeSse | ✓ | | | | ✓ |
| FIELDGEN | | ✓ | | ✓ | |
| GMS | ✓ | ✓ | ✓ | ✓ | |
| GSLIB | | ✓ | | | ✓ |
| GSTAT | | ✓ | ✓ | ✓ | |
| iTough2 | ✓ | ✓ | | | ✓ |
| Isatis | ✓ | | | | ✓ |
| MAD | ✓ | | ✓ | ✓ | |
| PEST | ✓ | | ✓ | ✓ | |
| SGeMS | ✓ | ✓ | | | ✓ |
| TPROGS | ✓ | | | | ✓ |
| spMC | | | ✓ | | ✓ |
| Isim3D | | ✓ | ✓ | ✓ | |

**Table 3: Properties of reviewed stochastic software solutions. Third-party GUI indicates that one or more Graphical User Interface solutions are available for the listed program, but implemented in a separate solution than the listed program.**

10   **4 A Comparative Study: Hydrogeology and Petroleum**

Several of the contributors to the 2004 and 2016 paper series have pointed to a lack of education in SH, along with stochastic analyses being too costly and not providing enough benefit, as main reasons for the lack of adoption of these methods in practice (Cirpka and Valocchi, 2016; Dagan, 2004; Fiori et al., 2016; Molz, 2004; Neuman, 2004; Sudicky, 2004; Winter, 2004). In response, we aim to compare the factors affecting the adoption of new methods in hydrogeology to those in



petroleum engineering, where stochastic methods for characterizing, mapping, and modeling the subsurface have become commonplace (e.g., Floris et al., 2001; Jonkman et al., 2000; Liu and Oliver, 2003; Oliver and Chen, 2011; Rwechungura et al., 2011). The factors we aim to compare include incentive structure, education, and the interaction between academic and industrial organizations.

**4.1 Incentive**

To compare the two industries, we start by recognizing the fundamental difference in the corporate incentive structure, which in turn leads to a difference in operations and adoption of new practices. In the petroleum industry, any improvement in subsurface characterization and mapping capabilities would ultimately lead to increase in profits, whether in the form of enhanced productivity or reduction in risk. Reservoir characterization and reserve estimates have a major impact on the

bottom-line of a company. They affect interactions with Wall Street (Misund and Osmundsen, 2015), and they are strictly regulated by the Securities and Exchange Commission (SEC). This relationship between petroleum engineering and the financial markets relies on guidelines set by both the SEC, as well as the Society for Petroleum Engineers, on how to communicate the certainty in the inherently uncertain reserve estimates (Harrell and Gardner, 2003). By having both professional and regulatory bodies dictating the use of probabilistic methods, the petroleum engineering field has significant

incentive to stay at the forefront of innovation and application regarding stochastic methods. In addition, the enormous capital costs associated with new and continued wellfield operations motivates the use of the most informative, risk-based predictions of costs versus benefits. In summary, petroleum firms are highly driven to create the most accurate depiction of underground resources in order to both accurately report reserve estimates and to best allocate costly drilling resources. Simply put, there is a direct link between improved characterization and improved profitability.

In hydrogeology, on the other hand, improvements in characterization and modeling do not always directly lead to improvements in profitability for firms involved with groundwater investigation and contaminant remediation. There are several factors that affect the perceived costs and benefits of adopting stochastic methods in hydrogeology, thus obfuscating the incentive structure. Very often, hydrogeologists involved in groundwater contamination projects must explain and justify steps taken, methods used, and conclusions reached to a wide range of people, including scientists in other fields, site

owners, regulators, attorneys, and the general public. In light of these factors, it may seem more justifiable to adhere to established, commonly-held concepts because adopting new methods could invite criticism. Furthermore, as will be elaborated upon in Sect. 5, it may often be the case that hydrogeologists are eager to use stochastic methods for uncertainty quantification, while their clients are reluctant to do so due to the perception that admitting any uncertainty may be construed as ignorance, thus leaving the client vulnerable to penalties from regulatory agencies.

There are no clear tangible financial benefits for improved modeling of uncertainty on the hydrogeology side. Environmental degradation, in simple economic terms, is a market failure and requires external intervention, e.g. regulations, to prevent. In




terms of innovation with respect to environmental protection, the incentive must come from regulation, which heretofore has been lacking (see Sect. 5). Consider, however, a (hypothetical) situation where owners of contaminated sites are held liable for not reporting accurately the financial liabilities implied by their contaminated sites, much the same way that energy companies are required to provide accurate estimates of their oil and gas reserves. Private companies would be required to

include such liabilities in their balance sheets whereas local and state governments would be required to set aside the resources needed for remediation. This would bring hydrogeology on par with petroleum engineering in the utilization of stochastic methods, since both would be equally motivated to find the oil, both metaphorically and literally. As things stand now, petroleum companies make money from finding oil. But that's not true for anyone else.

### 4.2 Education

Several contributions to the 2004 and 2016 series identified the lack of university courses in stochastic methods and, in consequence, a lack of hydrogeologists trained in stochastic methods as one of the setbacks to widespread adoption of stochastic methods in practice (Neuman, 2004; Winter, 2004). The lack of courses is driven, as explained in these two paper series, by what appears to be a high-level of math and abstraction and seemingly, lack of practicality. One way to investigate this hypothesis is to look at differences between the education of hydrogeologists and reservoir engineers, as it is reasonable

to expect that academic education would respond to the demands of industry.

Continuing the comparative analysis of these two industries, we compared the prevalence of courses in these subjects in the disciplines most likely to be studied by future petroleum engineers and practicing hydrogeologists. Three academic disciplines were selected for analysis: petroleum engineering, civil/environmental engineering, and earth sciences. We surveyed the top ten schools in the US News & World Report 2016 rankings for graduate programs in petroleum engineering

and earth sciences. Rankings for civil engineering are done separately from environmental engineering, and the top ten of the two rankings produced thirteen unique universities, which were the universities included for this survey.

Gathering information is made complicated because the amount of information about course offerings was variable from school to school. For some schools, course titles and course descriptions were available, while for others only course titles were available. For a couple programs, no course information was publicly available online. While rankings exist for "Earth

Sciences" as a subject, it is difficult to make direct comparisons due to the various organizational structures.

Our goal was to look for courses focused on stochastic methods in characterization and mapping. While utilization of geostatistical methods is not always necessary for incorporation of stochastic concepts, they are often used in conjunction, so our search included courses in geostatistics. For schools where course information was found, seven out of nine petroleum engineering schools had a course where geostatistics is the main focus (i.e. in the course title), and one school had a course

where geostatistics is mentioned as part of the course. On the other hand, four out of twelve of the civil/environmental programs had a course where geostatistical methods were the main focus. Two programs had a course that included





geostatistics or SH in the course description, but were seemingly not a main focus.  Only one Earth Sciences program had a course focusing on SH. Many civil/environmental engineering or Earth sciences programs have at least one course on the topics of probability, statistics, or uncertainty analysis in some regard, but not specifically related to hydrology.

Note that this survey does not include information about how often these courses are taught, or what percentage of students takes these courses.  This survey also does not account for the possibility that these courses are being offered in other departments and taken by students of the departments in question. Despite these limitations, the results are clear: courses in geostatistics and stochastic methods are much more prevalent in petroleum engineering departments than in civil/environmental engineering or earth sciences departments. This finding suggests to us that there is no fundamental pedagogical difficulty in teaching stochastic methods. The demand for classes is not there, and it is very likely curbed by the limited marketability of the SH skillset in groundwater-related industries. Class offerings are driven by demand and are rarely curbed by intellectual challenges, especially in top academic programs.  Of course, classroom teaching is just one mode of instruction and training, the other being research. This brings us to our next subsection, where we will take a closer look at this aspect of teaching.

## 4.3 Research Culture/Collaboration

The vast majority of research in any scientific field comes from institutions belonging to one of three categories—academic, governmental, and industrial.  Significant collaboration in research between these three sectors of the economy can act as a catalyst for adoption of novel methods, due to focusing of research goals, improved funding, and faster transfer of research-based knowledge from lab to field.

Since participation by industrial institutions in research is one avenue for adoption of new methods in practice, we aim to compare this activity in the two fields. To accomplish this, we performed a bibliometric survey using Web of Science similar to the one performed by Rajaram (2016). The goal of the survey was to quantify the extent of collaboration in research between academia, government, and industry in each of the two disciplines.

The bibliometric survey was performed in March 2018 by searching for all articles matching keywords related to stochastic methods in the two fields. The author affiliations of the resulting articles were recorded, and ordered by number of articles for each institution. All institutions that appeared on at least three articles were recorded and categorized as one of academic, governmental, or industrial. The number of appearances of each category was calculated for each of the two disciplines, with results presented in Figure 1. Details of the survey and processing are provided in Appendix A.

For both hydrogeology and petroleum engineering, the majority of affiliations are academic, which may be expected for any field, but the ratio of academic to industrial affiliations is approximately 211:1 for hydrogeology whereas it is only approximately 23:1 for petroleum engineering, a tenfold difference in industry involvement. Hydrogeology has approximately 10 times as many publications as petroleum engineering overall, but petroleum engineering still has more





industry appearances at 15 affiliations (e.g. PetroChina, DWA Energy Ltd., Chevron, and Schlumberger,) compared to hydrogeology's 12 industry appearances (e.g. Aquanty Inc., Schlumberger, and Tetra Tech Inc.). While this affiliation search is not completely exhaustive for all publications, languages, and possible search terms, the results are conclusive: there is clearly a larger presence of industrial institutions contributing to research in the petroleum field compared to hydrogeology.

These results confirm the hypothesis that there is much greater participation in research by industrial institutions in the petroleum field than in hydrogeology. In turn, it can be argued that this indicates a stronger connection between theoretical development and practice: advances in theory are both motivated by practice and smoothly adopted by practice. These observations are closely aligned with our earlier observations on course offerings in the respective academic programs, where we noted a much stronger emphasis on stochastic methods in petroleum-related schools, driven by a strong interest

from industry.

**4.4 Comparison Summary**

Despite being closely aligned in theoretical bases, hydrogeology and petroleum fields are fundamentally different in their incentive structures. The direct nature of the relationship between methodological improvements on one hand and increased profitability plus regulatory constraints on the other provides impetus for advancement in the petroleum industry, which

translates into funding opportunities and demand for skilled graduates. Schools are easy to adapt to market forces through course offerings and faculty recruitment. However, such relationships are not present in hydrogeology, leading to stagnation in industry and anemic interest in academia. When considering how often industry collaborates with academia, as quantified by journal article affiliations belonging to industry, the pattern remains: the petroleum engineering field has closer ties between industry and academia than the hydrogeology field does when it comes to SH research.





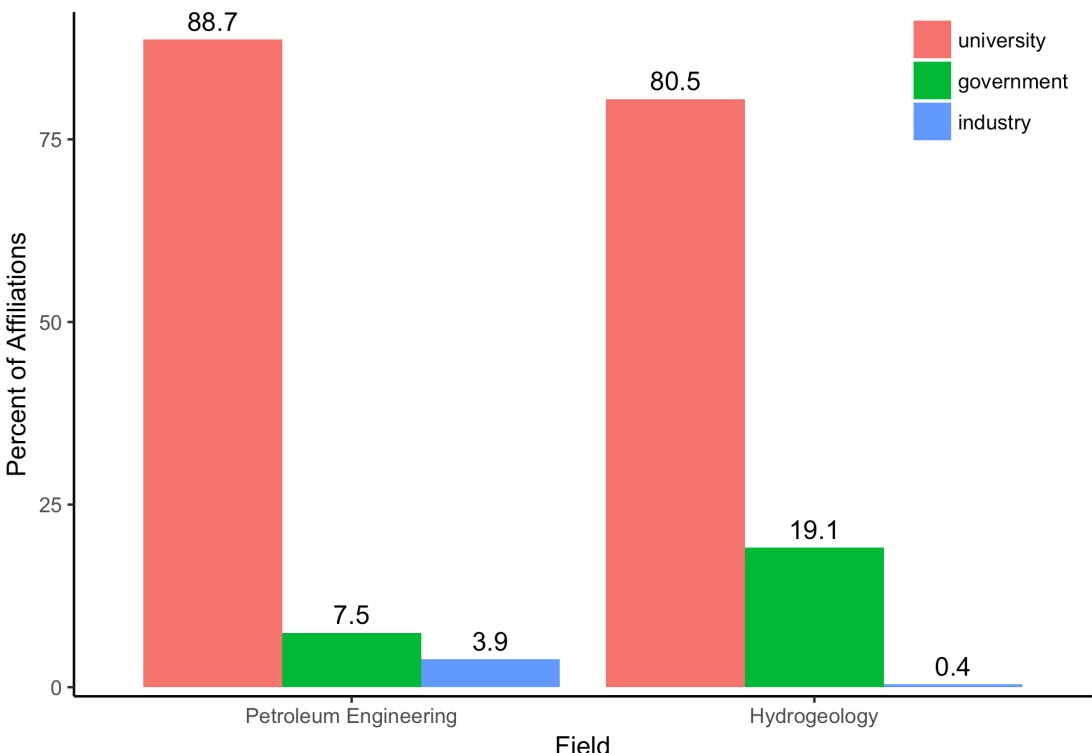

**Fig. 1: Bibliometric survey results for university, government, and industry author affiliations for petroleum engineering (left) and hydrogeology (right). The numbers on top of the columns indicate the percent of affiliations per category. The percent of affiliations reflect the proportion of total unique affiliations per document returned by the Web of Science database for each category (see the Appendix for full survey details). The total unique affiliation is a weighted sum of unique affiliations returned by Web of Science with weights being the number of documents containing that affiliation at least once, i.e. two authors from the same institution on the same document are only counted once.**

## 5 The Law, the Courts and Regulations

Previous publications opined that the regulatory and legal systems are not receptive to stochastic concepts and viewed this as one of the factors contributing to the gap. Early work on environmental regulations in the context of stochastic hydrology, e.g. Sudicky (2004), and more recent works, e.g. Fiori et al. (2016), noted that regulations do not require quantification of uncertainty, which would of course limit the demand for SH applications. Fogg and Zhang (2016) and Rajaram (2016) presented a somewhat different point of view and identified a drive towards a more explicit treatment of uncertainty.

### 5.1 Case Study – Risk-Based Corrective Actions

Some perspective on the regulators' approach for managing uncertainty could be gained by looking at the regulations associated with Risk-Based Corrective Actions (RBCA, US EPA (1989)) as a case study. RBCA focuses on cancer risk, defined as the upper bound lifetime probability of an individual developing cancer as a result of exposure to a particular level



of a potential carcinogen (US EPA, 1989). For example, following Smalley et al. (2000), a risk of $10^{-6}$ represents an increased probability of one in one million of developing cancer (De Barros et al., 2009; De Barros and Rubin, 2008; Maxwell et al., 1999). Assessment of the exposure risk requires analysts to consider all the elements contributing to this risk, which would include the source, contaminant transport pathways, exposure pathways, and toxicology parameters. Thus, by requesting and expecting risk assessment, specifically formulated in probabilities, RBCA specifically and explicitly recognizes the stochastic nature of making predictions under uncertainty. The rationale underlying RBCA is extended to other focus areas of the US Environmental Protection Agency (EPA), such as risk-based decision making in underground storage tank corrective action programs (US EPA, 1995). As a starting point for our conversation, we should note that risk-based assessments, and hence the underlying probabilistic approach, are endorsed by regulators.

**Fig. 2: EPA's letter encouraging (but not requiring!) the use of RBCA (US EPA, 1995). This document endorses a probability-based concept and refers to it as "sound science and common sense".**

The document shown in Figure 2 states that "A risk-based approach is consistent with the Administrator's efforts to ensure that our environmental cleanup programs are based on the application of sound science and common sense and are flexible and cost-effective". With the link between risk and probability clearly defined, such as in RBCA, the regulator's support of stochastic concepts is clear and compelling. A rationale similar to RBCA's was pursued in the development of protective risk-based levels for the contaminants subject to the EPA's Resource Conservation and Recovery Act (RCRA). What could possibly be confused to be a hindrance is the language (used, for example, in Directive 9610.17 in Figure 2 and US EPA





(1995)) that encourages the use of RBCAs but does not mandate them. This, however, would not be an accurate interpretation, as the EPA is a Federal agency, which puts it in a position to make recommendations and leave enforcement to states. EPA recommendations may be ignored by states, but this ignorance has consequences. Indeed, follow-up studies (e.g. Rifai and Suarez (2000)) confirm that most states in the US have in fact accepted the EPA recommendations and

formalized them as regulations, and that these regulations were enforced in a large number of field sites. With all this, it can be stated that regulations, if and when they exist, do not in principle pose a hindrance to application of stochastic concepts. With such a broad embrace of stochastic concepts by the EPA, one would expect a broad adoption of SH by practitioners. This obviously is not the case.

## 5.2 A view on the courts, the judiciary, and administrative agencies

Administrative agencies do not operate in a vacuum, but rather in a public policy continuum that also includes the judiciary and the legislation, and the politicians who connect them. It is the interaction between these institutions that dominates the attitude towards uncertainty and this is where agencies and courts get their cues. In this section, we seek to explore this continuum and assess the attitude towards probabilistic concepts and modeling of uncertainty. The main question we wish to address is whether SH has experienced limited adoption outside academia due to some structural constraints rooted in the

public policy continuum.

The view of the judiciary towards issues of uncertainty has been reviewed and heavily criticized by legal scholars, and the picture that emerges is that of inconsistency and confusion. There appears to be no fundamental difficulty with probabilistic concepts. There is no law, to our knowledge, that forbids presenting probabilistic concepts. Furthermore, there are several cases where courts considered and evaluated probabilities. Courts made some astute comments on probabilities. For

example, when reviewing a low-probability (18% chances of success!) plan proposed by a federal agency (NOAA Fisheries), the US Court of Appeals famously stated that "Only in Superman Comics' Bizarro world where reality is turned upside down, could the [National Marine Fisheries] Service reasonably conclude that a measure that is at least four times likely to fail as to succeed offers a 'fairly high level of confidence'" (see Owen (2009)). Currell et al. (2017), commenting on the legal system in Australia, noted that courts could deal with evidence in whatever form it is presented, and they may accept it,

reject it, misinterpret it, request clarifications and advice, ignore it, refer cases to other courts, or kick the can down the road. Rejection of probabilistic concepts in the judiciary, if and when it occurs, may be driven not by resistance to probabilistic concepts but possibly as a matter of convenience. One way of avoiding hard decisions, sometimes adopted by the courts, is relying on what is euphemistically called "adaptive management", which assumes that management will be able to respond to changes. Although Currell et al. (2017) refer to the Australian judiciary, their findings seem to apply also in the US.

At the same time, courts have been sending confusing messages. Farber (2011, p. 907) noted that "courts have sometimes reproved agencies for ignoring unquantified hazards, sometimes given their blessing as agencies buried their heads in the





sand, and sometimes even forbidden agencies to act in the absence of quantification". This inconsistency leads to ambiguity, and it is reasonable to expect that professionals may want to avoid this uncharted territory. What are the root causes of this ambiguity? Legal scholars see three main factors contributing to this ambiguity: congress, politicians and the lack of probability standards. Although we present them as separate items, they are obviously interrelated.

The US Congress underestimated and ignored environmental uncertainties because "Uncertainty management may seem a 'politically unappealing topic'" (Owen, 2009). European legislature is even less prescriptive, opting for the very vague precautionary principle as a guideline. To understand Congress, we need to look at politicians. In the absence of clear guidance from the legislature, agencies opted for non-binding guidance. A few environmental statutes contain signals about managing uncertainty, but these signals are often vague and indirect, leaving agencies with minimal guidance and a
significant amount of discretion (Owen, 2009). To fill in the void left by Congress, agencies have rarely promulgated rules or set standards for dealing with uncertainty. More frequently, they opted for providing non-binding guidance, which, as stated by Owen (2009), has very little visibility and often leaves major issues unresolved.

Politicians prefer to avoid stating the uncertainty associated with the stated goals of their proposed legislation and "it probably is unrealistic to expect legislators to candidly acknowledge the reality that their prescribed planning approaches
will not always work out" (Owen, 2009). The attitudes by politicians are also driven by the public's view of uncertainty being a consequence of disagreement between scientists and science disciplines, and as such might as well be ignored.

The third factor to consider is the lack of *probability standards*. Probability standards are defined by Owen (2009) as "standards addressing plans' chances of achieving mandated outcomes". The lack of probability standards is, for example, what allowed NOAA Fisheries to submit a plan with 18% probability of success (Owen, 2009). The lack of probability
standards is possibly the most critical of the three factors, because such standards, have they existed, would promote "more systematic and transparent confrontations with the uncertainties inherent in environmental planning" and would instill a "somewhat more cautious approaches to management of those uncertainties" and "reduce the persistent gaps between written environmental standards and actual performance" (Owen, 2009).

The negative consequences due to the lack of probability standards are numerous. It enhances risk-taking. It encourages
making decisions on an ad-hoc basis, which means that once a particular site is chosen, political pressure naturally will build to make the evaluative system fit the site rather than the underlying public safety goal" (Owen, 2009). It encourages individualized judgment and improvisation. The lack of probability standards "can turn a stringent protection standards into a perpetually unattained aspiration, lowering short-term compliance costs but also creating persistent slippage between legal requirements and actual performance" (Owen, 2009). Furthermore, without clear probability standards, assessments
undertaken with different methodologies lead to different conclusions. Such challenges are particularly difficult to address when several regulatory agencies are involved, with disparate resources and levels of expertise (Rothstein et al., 2006). It



resulted in "a patchwork response to uncertainties about plan success", and finally, the lack of probability standards closes the door for SH analyses and applications.

There are plenty of reasons why probability standards are avoided, a few justified and others opportunistic, but in any case they are not related to strengths or weaknesses of statistical theories. First, it is often argued that setting standards would limit the dialog, and hence the latitude for political maneuvering, on the tradeoffs between level of protection on one hand, and compliance costs on the other. Probability standards therefore have substantial environmental and economic implications.

As already alluded to, there is the "All Politics is Local" factor (O'Neill and Hymel, 1993). The Interstate Technology & Regulatory Council (ITRC) (2005) points out that "Risk-based decisions are sometimes met with intense and divisive criticism by industry and environmental stakeholders, as well as the public" due to normative conflicts concerning how risks of adverse events and the associated impacts are viewed technically, socially, and politically by the affected constituencies. National or international regulatory standards may not be accepted by local communities or businesses if those actors assess and balance costs and benefits in different ways. Some stakeholders see risk-based approaches as residing in the "world of specialists" and essentially closing off debate about broader historic, social, and economic issues (Rothstein et al., 2006). Regulators, civic groups, and business interests apparently prefer to settle environmental issues without being constrained by federal "interventionist" policies. In addition, there is the issue of financial incentives. As noted in Bullard and Johnson (2000), the current regulatory environment "creates an industry around risk-assessment and risk management", lmplying a satisfying status-quo for some of the stakeholders, and hence a lack of financial incentives for change by the professional community. And this occurs despite "faulty assumptions in calculating, assessing, and managing risk".

The case in support of setting probability standards is not without its criticism (Rothstein et al., 2006), but it may also have the unintended consequence of focusing policy attention on problems that carry high institutional risks for the regulator at the expense of those that carry high risks to society. Against this background, it does not seem that setting probability standards is imminent. The courts acceptance, or not, of probabilistic concepts is perhaps not the key question to ask in this context. A more fundamental question, dated possibly to the age of Galileo, is whether courts should be the arbiters of good science. A thorough review of the interaction between courts and groundwater hydrologists is provided by Currell et al. (2017). They concluded that, whereas courts are not prevented from dealing with uncertainty, they are not well-equipped to do so. Hydrogeologists should not expect courts to take an activist approach for promoting SH methods. This is not within their purview. Unfortunately, professional hydrologists are generally following the legal theories set by attorneys. Litigators may not be receptive to modeling of uncertainty because uncertainty means ambiguity, which may translate into unfavorable judgment. Attorneys claim that their clients are innocent, not "may be innocent". And similarly, prosecutors declare the defendants to be guilty, not "maybe guilty". In other words: "uncertainty? What uncertainty? There is no uncertainty!", and so, who needs SH?



Realizing that courts should not be the arbiters of science, an important outcome from the extensive Australian experience in litigating groundwater hydrology Currell et al., 2017), has been the establishment of independent expert scientific committees to advise the courts in all matters scientific. An example where such involvement opened the door for SH can be found in the Environmental Radiation Protection Standards for Management and Disposal of Spent Nuclear Fuel and

Transuranic Radioactive Wastes (40 CFR Part 190). The standards require applicants to perform a probabilistic assessment of long-term risk to the public due to accidental releases. An example of analyses based on these Standards is provided in Zimmerman et al. (1998). This publication demonstrates that clear guidelines and quick responses are certainly possible. Why then such a difference between nuclear waste situations and what could possibly be viewed as the "lesser evil"? Legal scholars associate this special case to with the involvement of a high-caliber scientific board in the form of the National

Academy of Sciences (Farber, 2011; Owen, 2009).

In summary: Courts are not prohibited from dealing with probability, but they have sent confusing messages with inconsistent rulings. Agencies lack probability standards for addressing their "plans' chances of achieving mandated outcomes". The guidance provided in lieu of regulations is non-binding and fraught with problems. Congress, in the US, underestimated and ignored environmental uncertainties. European countries rely in the precautionary principle when

dealing with uncertainty, which is vague and it is even less prescriptive. Politicians refrain or hesitate acknowledging that their plans are associated with uncertainty. And as a consequence, SH is facing formidable headwinds on its way to become applicable, and this has nothing to do with the strength of the science. Unfortunately, there is no indication of any dynamics that will pull SH into a more favorable position, at least not from this angle.

## 6 The BIG Miss – SH Blind Spot

Previous sections explored the challenges facing SH on its road to becoming broadly accepted by practitioners, regulators, and administrators. These challenges do not reflect weaknesses in the underlying SH science. There is, however, at least one challenge that is real and urgent, that has been around for a long time and that has been ignored by SH to its own detriment. This issue can be described by quoting former US Secretary of Defense Donald Rumsfeld who, when addressing the challenges of making decision under uncertainty, famously distinguished between known knowns, known unknowns, and

unknown unknowns. SH has been dealing primarily with the known unknowns and ignored the most challenging part of uncertainty management which covers the unknown unknowns.

What are known unknowns? As explained by Rumsfeld, known unknowns are things we know that we do not know. In hydrogeology, this would include examples such as the spatial distributions of the hydraulic conductivity and almost any other soil property and, by way of consequence, state variables such as hydraulic head, concentrations, and solute travel

times. It can also include forcing terms such as recharge (see Rubin and Dagan (1987a, 1987b)).



What did Rumsfeld mean by "unknown unknowns"? Following Rumsfeld's definitions, these are the things we do not know that we do not know. These are the situations where a variable is intentionally or unintentionally ignored. An example for this can be drawn from the case of nuclear waste disposal. What makes this problem unique is the life span of compounds such as plutonium. With a life span of 250,000 years, the list of unknown unknowns could include, for example, climate

change and possible changes in human behavior, societal norms, and political systems. Farber (2011) noted in this context that "the calculation of the physical likelihood of leakage should (hopefully) be reasonably accurate, the assumptions about human presence and activities in the area (and therefore about exposure) are speculative, and the likelihood of human interference with the integrity of the site is completely unknown". Recalling the adaptive management concept that is sometimes employed for dealing with uncertainty, Owen (2009) noted that given the time span of many hydrological

processes, "One therefore cannot reasonably assume that present social or political institutions will survive long enough to adjust waste containment systems", and thus rendering unreasonable any semblance of planning.

In the case of nuclear waste, as in almost any case, it is hard to fathom predicting 250,000 years down the road. For some perspective, consider that the pyramids were built only 5000 years ago. The ancient Egyptians did a good job at hiding the mummies and treasures for thousands of years, but who could have predicted ground-penetrating radar. Long-term

prediction is not a simple question, but possibly the most relevant question. A confirmation that the unknown unknowns are of topmost significance for regulators is given in National Research Council (2014). This publication, dealing with risk-informed decision-making in contaminated sites, identified the unknown unknowns, i.e. the drivers of uncertainty, as the major and yet unanswered challenges in the decision-making process, including, for example, (1) "uncertainties that contamination may exist at a site", (2) estimating the "quantities of contamination that may exist in the future" (p. 91), (3)

"low probability, high consequence events" (p. 92; authors' note: these are NRC's code words for human-related scenarios), and (4) uncertainty about how to predict failures of treatment and containment systems (p. 123).

Yucca Mountain, a central activity for hydrogeologists for more than 40 years, is a paramount example of SH failure. How did SH address the uncertainty challenges? As noted earlier, Farber (2011) concluded that "calculation of the physical likelihood of leakage should (hopefully) be reasonably accurate" (our note: covered by SH, see Zimmerman et al., 1998),

"the assumptions about human presence and activities in the area (and therefore about exposure) are speculative" (our note: not covered by SH), "and the likelihood of human interference with the integrity of the site is completely unknown" (our note: not covered by SH). Notably, there was no reference to issues such as modeling of contaminant transport as being showstoppers. SH opted to dealing with the physics of flow (known unknowns, using Rumsfeld's terminology and otherwise known as "risk analysis' blind spot" using Farber's terminology), but ignored the societal risks residing on the interface

between physics and the human environment (unknown unknowns, otherwise known as uncertainty). This omission did not escape the attention of the courts: in the absence of appropriate tools, the site planners chose to ignore "human activities and





economic imperatives", opting instead to assume current conditions, a decision which was referred to by the court as "odd" (Farber, 2011, p. 951).

Modeling the unknown unknowns is a daunting task, but ignoring it has grave consequences, as Rumsfeld himself attested. It also seems that ignoring it is not an option, as the legal stance is pretty clear. Farber (2011, p. 910) quotes the court as
saying that "the inability to quantify a risk does not justify failure to discuss it if there are other grounds for considering it significant".

Returning to our energy industry comparisons, petroleum engineering does not face such issues because most projects are short duration and are well-defined by scientific and engineering considerations, with minimal or no interfacing with non-engineering issues. But petroleum engineering also fails in situations where human factors are involved, such as in the case
of British Petroleum's Deepwater Horizon well failure and oil spill in the Gulf of Mexico and also the now widely refuted peak oil theory.

Having therefore outlined the challenge, what sorts of solutions have been developed in the field of uncertainty characterization and uncertainty reasoning for dealing with unknown unknowns and can these solutions be transferred to the field of hydrogeology? An important milestone here is the groundbreaking work of Savage (1954), which provided
probability theory with the necessary link to decision theory. So strong was the impact of Savage's work that it not only became the mainstay of decision theory everywhere (Baron 2004; Berger, 1985; Bernardo and Smith, 1994; Gilboa, 2009; Jeffrey, 1992; Kahneman et al., 1982; Peterson, 2009)), that even when Nobel laureates (Kahneman and Tversky, 1973) showed how decision making in real-life often diverge from the axioms described by Savage (1954), their ideas were largely ignored (initially). On the contrary, probability theory and probabilistic decision theory was simply declared to be the
rational norm and anything else is irrational.

This harsh repudiation of people's intuition began to slowly change with the growing recognition that real-world situations are ripe with unknown unknowns due to ignorance, ambivalence, or ambiguity. In such situations, it is not easy to see how a unique prior distribution can be defined by an agent as demanded by classic decision theory. To account for unknown unknowns, two lines of thought can be identified. The first approach stays within the well-defined confines of probability
theory and accounts for possible unknown unknowns by using a whole set of prior distributions instead of just a single unique one. Updating of beliefs is then performed by Bayes' rule and the common rules of decision theory, i.e. the maximization of expected utility (see, e.g. Berger (1990); Wasserman and Kadane (1990)). The second approach goes beyond probability and is based on the works of Dempster (1968) and Shafer (1976). Here, beliefs are no longer represented by a probability function but by a so-called belief function, and for updating, Bayes' rule is replaced by the Dempster-Shafer
rule. This Dempster-Shafer theory has historically been criticized for its failure to connect with decision theory (Pearl, 1990), but more recent developments by Gilboa and Schmeidler (1993) have provided solid foundations for deriving decision rules from Dempster-Shafer belief functions, too.



Whereas we think that in science alone, SH included, the importance of unknown unknowns is constantly decreasing with the accumulation of knowledge through research, the prediction of societal, political, economic, technological, and demographic changes will remain a source of unknown unknowns for the foreseeable future. Since scientific results constantly interface with these fields through the perspective of decision makers, we should also expand the applicability of

SH into this domain.

**7 Summary**

Several publications attempted to explain the apparent failure of SH to gain broad acceptance in application. In this paper, we reviewed the various hypotheses raised and evaluated them by establishing and analyzing a factual basis. We found a large disparity between several of the hypotheses and the facts on the ground as we see them. For example, we show that it is

not justified to view SH education or software as roadblocks on the way to broad acceptance.

The roadblocks on the way to making SH broadly accepted are primarily two. One roadblock is external, meaning that it is not related to SH fundamentals. The other factor is in fact internal as it is related to SH self-definition: the topics it chose to pursue, and more important, those that it chose, knowingly or not, to ignore:  it is what we call SH's blind spot, and it is related to SH fundamentals.

The external factor resides in the continuum defined by the legislature, the judiciary, and government. Legal scholars summarize it succinctly as a lack of probability standards. Probability standards are defined by Owen (2009) as "standards addressing plans' chances of achieving mandated outcomes" (see discussion in Sect. 5). The lack of probability standards institutionalizes the fragmentation between the three branches of government often asked to deal with uncertainty and eliminates the financial incentive needed for industry to move in and address the challenges.

The internal roadblock is the focus on risk (Rumsfeld's known unknowns) and ignoring uncertainty (the unknown unknowns). Uncertainty resides on the interface between hydrogeology and related disciplines such as climate science and sociology/political science. It is noted that Yucca Mountain project failed because of uncertainty; not because of any doubt about the values of the conductivity anywhere, but because of inability to make long-term predictions regarding climate change and to model the possible changes in human behavior, societal norms, and political systems. These are by no means

easy challenges, but work on these topics is already under way, alas, not by hydrologists. Expanding SH to account for risk as well as uncertainty is a key for making an impact. This lacuna is not the privilege of SH: as noted by several authors, there are other environmental problems needing to address uncertainty, such as fisheries, air quality, endangered species, water quality, and climate change (see Owen (2009)). And of course outside the environmental sciences we have financial risks and other catastrophes. Publications on the interface between environmental science and the social sciences are available

(Liu et al., 2007; Torn and Harte, 2006) and quite a few from legal scholars (reviewed earlier), but SH is still lagging.



## Appendix A

The bibliometric survey comparing the collaboration between academia and industry for the two fields of hydrogeology and petroleum engineering consisted of first compiling information from the Web of Science database at http://apps.webofknowledge.com/ (v.5.27.2 during March 2018) and then processing that information into useable metrics. This database allows users to first query certain search terms to generate a list of documents and then access summaries about that list such as the number of articles published per year or how often organizations are included in the author affiliations. To perform the analysis of comparing the collaboration of academia and industry in the two fields of interest, further data summary metrics were calculated with the results downloaded from the Web of Science. The two lists below detail the steps taken for these two procedures.

Steps for compiling data from Web of Science:

1. Search terms were defined to capture relevant documents for the two fields of hydrogeology and petroleum engineering. See Table 1 for the search terms used. There were five sets of iterative search terms used per field to confirm that the resulting metrics comparing the two fields were consistent.

2. Each of the search term sets were queried with the Web of Science database using the advanced search option. A time span of the last five years (2013-2018) was selected, and all documents types were used, i.e. the search was not limited to just journal articles in order to capture collaborations in conference posters and presentations or other documents.

3. For each search set, the Analyze Results feature was used to generate information regarding the affiliations listed for the documents returned by the query. The Organizations field was selected with a minimum record threshold set to 3, i.e. an organization had to be listed on at least 3 documents, in order to make the volume of unique affiliations manageable for manual classification. The data returned include organization names and both the raw number and percent of articles in the search containing those organizations as an affiliation. These numbers reflect how many documents in which the organization is listed as an affiliation for at least one author. The results table was downloaded as a text file for each search set.

Steps for calculating metrics:

1. Each organization returned by Web of Science was manually categorized as a degree-granting academic institution ('university'), a government agency or research organization primarily funded by government agencies ('government'), or, if neither of those, then 'industry'. This was accomplished by searching for certain text patterns in the organizations' names. For example, if the pattern "univ" or "college" was in the organization name, then it was placed in the 'university' category. Or if "geological survey" or "national research council" was in the organization name, then it was categorized as 'government'. Additional patterns were created until all organizations were categorized. When the name of the organization was not sufficient to assign a category, the organization was investigated to ascertain the most appropriate affiliation.



2. For each field, category, and search set, the number of records across organizations were summed and a percentage was calculated. This percentage represents the proportion of affiliations on documents belonging to the three categories.

3. The relative relationship between the three categories per field was confirmed to remain consistent across the iterative search terms. The fifth set was used for the rest of the analysis.

4. The categorical percentages were compared between the two fields to ascertain if the two fields of hydrogeology and petroleum engineering had more participation of industry in research collaborations leading to publications or conference abstracts.

**Table A1: Web of Science search terms used to compare hydrogeology and petroleum engineering documents. The stochastic terms (leftmost column) were used for all search sets. For hydrogeology, the searches were limited to the Web of Science categories listed in the second leftmost column. The five search sets for hydrogeology are listed in the middle column and are composed of the search terms listed. The search sets vary by incrementally more search terms related to hydrogeology. Similarly, the Web of Science categories and search term sets for petroleum engineering are listed in the two rightmost columns. Each search set consists of combining the stochastic terms, the Web of Science categories, and field terms with a logical AND operator via the Web of Science. Note: the asterisk \* denotes the wildcard search feature, e.g. geostatist\* matches geostatistics and geostatistical.**

| | Hydrogeology | | Petroleum Engineering | |
|---|---|---|---|---|
| **Stochastic terms** | **Web of Science categories** | **Field terms** | **Web of Science categories** | **Field terms** |
| stochastic OR geostatist* | Water Resources OR Environmental Sciences OR Limnology OR Engineering Environmental OR Science | 1: porous OR soil OR aquifer OR subsurface OR hydrogeology OR geology | Petroleum OR Energy | 1: porous OR soil OR subsurface OR geology OR reservoir OR oil |
| | | 2: 1 + OR groundwater | | 2: 1 + OR petrol* |
| | | 3: 2 + OR remediation | | 3: 2 + OR exploration |
| | | 4: 3 + OR hyporheic | | 4: 3 + OR natural gas |
| | | 5: 4 + OR phreatic | | 5: 4 + OR hydraulic fracturing |



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
