# Peer review of "Stochastic Hydrogeology's Biggest Hurdles Analyzed and Its Big Blind Spot"

_Hydrology and Earth System Sciences, 2018_

## Referee Comment (RC1) · Anonymous Referee #1 · 23 Jun 2018

I have enjoyed very much reading the views of the authors about stochastic hydrogeology, its past, and its potential future. However, the paper is an opinion paper, not a research one.

---

## Referee Comment (RC2) · Anonymous Referee #2 · 23 Aug 2018

This paper considers the impediments to the adoption of stochastic methods in hydrogeology. The authors debate on very interesting questions arguing with very deep knowledge of the topic. I was very caught by this manuscript and their conclusions regarding the major roadblock - i.e. "the focus on risk (Rumsfeld's known unknowns) and ignoring uncertainty (the unknown unknowns)" - to making Stochastic Hydrogeology broadly accepted, have opened my mind. I don't know what is the editorial location of this manuscript but in my opinion it deserve to be published even in this form.

---

## Author Comment (AC1) · 10 Sep 2018

Thanks for your kind review. Much appreciated.

———————————————

---

## Author Comment (AC2) · 10 Sep 2018

Thank you for your review and we are glad you enjoyed our paper. Our paper started off with a review of two previous opinion/review paper series, but the intent was to dive deeper into the question of why stochastic methods are hardly adopted by practitioners. Beyond merely stating our opinions and providing a handful of examples, we sought to provide the evidence to paint a thorough picture of the situation from multiple angles - a feat that required research. We researched, categorized, and described a plethora of software packages to assess the situation of software availability. We researched higher education programs and their instructional support for stochastic methods. We researched bibliometrics statistics to quantify research collaboration patterns. We researched legal opinions on uncertainty in environmental regulation. Due to the multiple

angles that we pursued, and more importantly the depth into each, we respectively argue that this paper is a research article. In the descriptions of HESS manuscript types, it states that research articles "report on original research which clearly advances our understanding of hydrological processes and systems, and/or their role in water resources management" - we contend that, although our paper is not a traditional scientific research paper, it does advance our understanding of what roadblocks there are in the role of hydrological research in water resources management.